# Advancing Spinal Cord Injury Treatment through Stem Cell Therapy: A Comprehensive Review of Cell Types, Challenges, and Emerging Technologies in Regenerative Medicine

**DOI:** 10.3390/ijms241814349

**Published:** 2023-09-20

**Authors:** Chih-Wei Zeng

**Affiliations:** 1Department of Molecular Biology, University of Texas Southwestern Medical Center, Dallas, TX 75390, USA; chih-wei.zeng@utsouthwestern.edu; 2Hamon Center for Regenerative Science and Medicine, University of Texas Southwestern Medical Center, Dallas, TX 75390, USA

**Keywords:** spinal cord injury, stem cell therapy, neural tissue repair, embryonic stem cells, induced pluripotent stem cells, mesenchymal stem cells, regenerative medicine

## Abstract

Spinal cord injuries (SCIs) can lead to significant neurological deficits and lifelong disability, with far-reaching physical, psychological, and economic consequences for affected individuals and their families. Current treatments for SCIs are limited in their ability to restore function, and there is a pressing need for innovative therapeutic approaches. Stem cell therapy has emerged as a promising strategy to promote the regeneration and repair of damaged neural tissue following SCIs. This review article comprehensively discusses the potential of different stem cell types, such as embryonic stem cells (ESCs), induced pluripotent stem cells (iPSCs), mesenchymal stem cells (MSCs), and neural stem/progenitor cells (NSPCs), in SCI treatment. We provide an in-depth analysis of the unique advantages and challenges associated with each stem cell type, as well as the latest advancements in the field. Furthermore, we address the critical challenges faced in stem cell therapy for SCIs, including safety concerns, ethical considerations, standardization of protocols, optimization of transplantation parameters, and the development of effective outcome measures. We also discuss the integration of novel technologies such as gene editing, biomaterials, and tissue engineering to enhance the therapeutic potential of stem cells. The article concludes by emphasizing the importance of collaborative efforts among various stakeholders in the scientific community, including researchers, clinicians, bioengineers, industry partners, and patients, to overcome these challenges and realize the full potential of stem cell therapy for SCI patients. By fostering such collaborations and advancing our understanding of stem cell biology and regenerative medicine, we can pave the way for the development of groundbreaking therapies that improve the lives of those affected by SCIs.

## 1. Introduction

Spinal cord injuries (SCIs) are among the most debilitating and life-altering medical conditions, leading to a significant decline in the quality of life and imposing immense physical, psychological, and economic burdens on affected individuals and their families [1]. According to the most recent epidemiological data, the incidence and prevalence of SCIs vary significantly across different regions and demographics, emphasizing the widespread impact of this condition (Table 1). Conventional treatments for SCIs have been limited in their ability to promote meaningful functional recovery [2]. However, stem cell therapy has emerged as a promising avenue to address these limitations, with the potential to transform the landscape of SCI treatment. Recent studies have demonstrated the therapeutic potential of stem cells in treating SCIs while acknowledging the inherent challenges and obstacles that must be overcome [3,4]. In this review article, we explore the promise of stem cell therapy in the context of SCIs, the challenges that researchers face, and the importance of continued research to fully realize the potential of this groundbreaking approach.

Stem cell therapy offers hope for patients suffering from SCI by harnessing the unique regenerative capabilities of stem cells [16]. These cells have the potential to differentiate into various cell types [17], thereby replacing lost neurons, promoting axonal growth, remyelinating damaged axons, modulating immune response, and creating a permissive environment for functional recovery [18,19,20]. By targeting multiple aspects of the SCI pathology, stem cell therapy could overcome the limitations of traditional treatments and significantly improve the lives of affected individuals [21]. Several types of stem cells are currently being investigated for their potential in SCI treatment, including embryonic stem cells (ESCs), induced pluripotent stem cells (iPSCs), mesenchymal stem cells (MSCs), and neural stem/progenitor cells (NSPCs) [22,23,24,25]. Each of these cell types offers unique advantages and challenges in the context of SCI therapy. ESCs, for instance, are pluripotent and can differentiate into any cell type, but their use raises ethical concerns related to embryo destruction [26]. iPSCs, on the other hand, can be generated from adult cells, avoiding the ethical issues associated with ESCs, but they carry a risk of tumor formation [27,28]. MSCs are known for their immunomodulatory properties and have shown promise in preclinical studies [29], while NSPCs can be derived from various sources and have the inherent ability to generate neural cells [30]. Despite the tremendous potential of stem cell therapy for SCI treatment, researchers still face significant challenges that must be addressed before these therapies can be widely adopted. Ensuring the safety and efficacy of stem cell-based therapies is paramount. For example, the risk of tumor formation, potential for immune rejection, and the need for standardized protocols for stem cell isolation, expansion, and differentiation must be carefully addressed. Additionally, the development of suitable biomaterials and scaffolds to support cell survival, integration, and function is crucial for the successful translation of stem cell therapies from the laboratory to the clinic [31,32,33].

The use of human stem cells, particularly ESCs, raises important ethical concerns [34]. Transparent and robust regulatory frameworks are essential to ensure the safe and ethical application of stem cell therapies for SCIs [35]. Public dialogue and collaboration between researchers, clinicians, policymakers, and society at large are vital for addressing these ethical issues and developing appropriate guidelines. In conclusion, stem cell therapy offers a beacon of hope for patients suffering from SCIs and has the potential to revolutionize treatment by promoting the regeneration and repair of damaged neural tissue. As researchers continue to explore and refine stem cell therapies, we can expect breakthroughs that will transform the way we approach SCI treatment. By overcoming the challenges and addressing ethical concerns, stem cell therapy has the potential to make a significant impact on the lives of those affected by SCIs.

## 2. The Promise of Stem Cell Therapy for SCIs

Stem cell therapy has the potential to revolutionize the treatment of SCIs by promoting regeneration and repair of damaged neural tissue. By using stem cells, researchers aim to replace lost neurons [36], remyelinate axons [37], modulate the immune response [38], and create a permissive environment for axonal growth and functional recovery [39]. This approach could provide new hope for patients suffering from SCIs, who currently have limited treatment options. Additionally, stem cell therapy can target various aspects of SCI pathology, such as inflammation, apoptosis, and glial scarring [40,41,42]. By addressing these issues, stem cells may mitigate secondary damage and further improve the prospects for neural repair and functional recovery [43]. The versatility of stem cells and their ability to adapt to the specific needs of the injured spinal cord make them ideal candidates for developing tailored therapies that target individual patient requirements.

Recent preclinical studies have shown promising results in animal models of SCI, with stem cell transplantation leading to improvements in locomotor function, reduction in lesion size, and enhanced neural regeneration [44]. These findings highlight the potential of stem cell therapy to translate into clinically meaningful outcomes for SCI patients. Furthermore, ongoing clinical trials are evaluating the safety and efficacy of stem cell-based therapies in human patients with SCIs [45]. These trials represent a critical step toward the development of novel, effective, and safe treatments for individuals suffering from SCIs. As research continues to advance, the integration of novel technologies such as gene editing [22], 3D bioprinting [46,47], and tissue engineering [46,48] may further enhance the potential of stem cell therapy for SCI treatment. These innovations can help optimize stem cell delivery, survival, and integration into the host tissue, ultimately leading to more effective therapies and improved outcomes. Additionally, it is important to note that ongoing clinical trials are actively assessing the safety and effectiveness of stem cell therapies in human patients with SCIs [45]. These clinical trials are a crucial advancement in the field, paving the way for new, effective, and safe treatment options for those affected by SCIs. While the outcomes of these clinical studies have been mixed, they are generally promising. Some studies have even reported improvements in lost motor, sensory, and autonomic nervous system functions, underscoring the potential of stem cell therapies [49,50,51,52]. While it is true that the majority of stem cells differentiate into glial cells rather than neurons, these cells still play crucial roles in structural support, inflammation modulation, and remyelination, all of which contribute to the overall therapeutic effect [53]. Ongoing research is focused on optimizing conditions for neuronal differentiation and integration into existing neural circuits. This research is facilitated by technological advances such as gene editing and 3D bioprinting. For instance, fast-scalable 3D bioprinting has been shown to create biocompatible and biomimetic scaffolds that precisely fit the geometries of spinal cord lesions. These scaffolds promote axonal regeneration and support stem cell grafts, thereby aiding in the recovery from SCIs in rodent models [54]. Despite the bias towards glial cell differentiation, preclinical studies have reported improvements in both locomotor function and lesion size reduction.

### 2.1. Comparative Analysis of Stem Cell Transplantation and Other Cell-Based Therapies in SCIs

To provide a more comprehensive understanding of the therapeutic options available for SCIs, it is important to compare the advantages and disadvantages of stem cell transplantation with other cell-based therapies such as Schwann cells and olfactory ensheathing glia cells. Below, we outline the key points of this comparison.

#### 2.1.1. Advantages of Stem Cells

Stem cells offer the unique advantage of pluripotency or multipotency, allowing them to differentiate into various cell types, including neurons and glial cells. This makes them versatile candidates for addressing multiple aspects of SCI pathology, such as inflammation, apoptosis, and glial scarring [36,37,38,55].

#### 2.1.2. Disadvantages of Stem Cells

The use of stem cells comes with challenges such as the risk of tumor formation, potential for immune rejection, and ethical concerns, especially in the case of embryonic stem cells [26,27,28].

#### 2.1.3. Advantages of Schwann Cells and Olfactory Ensheathing Glia Cells

These cells are already specialized for roles in the nervous system, potentially reducing the risk of unwanted differentiation. Schwann cells are known for their ability to promote axonal growth, while olfactory ensheathing cells can remyelinate axons and modulate inflammation [50,51].

#### 2.1.4. Disadvantages of Schwann Cells and Olfactory Ensheathing Glia Cells

These cells are more limited in their differentiation potential compared to stem cells, which may restrict their ability to address the multifaceted pathology of SCIs [52].

In summary, stem cell therapy offers a promising and innovative approach to treating SCIs, addressing various aspects of the injury and providing the potential for significant functional recovery [56]. As research continues to progress and collaborations between researchers, clinicians, and patients strengthen, the full potential of stem cell therapy in the treatment of SCIs may soon be realized, offering renewed hope to those affected by these devastating injuries.

### 2.2. Biomaterials in 3D Stem Cell Constructs for SCI

The field of regenerative medicine has seen significant advancements in the development of biomaterials that can effectively hold 3D tissue-engineered constructs of stem cells. These biomaterials not only serve as scaffolds but also influence stem cell behavior, differentiation, and integration into host tissues. This section explores various types of biomaterials that have shown promise in this regard, including hydrogels, poly(ethylene glycol)-fibrinogen hydrogels, nanomaterials, and microgels.

#### 2.2.1. Hydrogels

Hydrogels such as spatially varying multi-layered hydrogels have been shown to stimulate efficient regeneration of complex tissues from a single stem cell population [57]. Another study demonstrated that hydrogels could be used as 3D cell carriers for the mechanical stimulation of encapsulated stem cells [58].

#### 2.2.2. Poly(ethylene glycol)-Fibrinogen Hydrogels

These hydrogels have been used to directly differentiate human pluripotent stem cells into contracting heart tissues [59].

#### 2.2.3. Nanomaterials

The combination of stem cells and nanomaterials is expected to develop into an important tool in tissue engineering [60].

#### 2.2.4. Microgels

Stem cell-laden microgels with shape-forming properties can be used as smart building blocks for tissue regeneration [61].

In summary, various types of biomaterials, including hydrogels, poly(ethylene glycol)-fibrinogen hydrogels, nanomaterials, and microgels, have shown promise in holding 3D tissue-engineered constructs of stem cells. These biomaterials offer unique advantages in terms of mechanical properties, biocompatibility, and the ability to influence stem cell behavior. Continued research in this area is essential for optimizing these biomaterials for clinical applications in regenerative medicine.

## 3. The Potential of Different Stem Cell Types

Various types of stem cells are being explored for their potential in SCI treatment. In this review article, we mainly discuss the four cell types, ESCs, iPSCs, MSCs, and NSPCs, which each offer unique advantages and challenges in the context of SCI therapy. By understanding the characteristics and capabilities of each cell type, researchers can optimize their use in regenerative medicine applications. We provide a comparison of the advantages and challenges associated with different stem cell types in the context of SCI treatment (Table 2). In this review, we delve into the therapeutic potential of various stem cell types—ESCs, iPSCs, MSCs, and NSPCs—in the context of SCI treatment. Each stem cell type presents its own set of advantages and limitations. For instance, ESCs offer a broad differentiation spectrum but are fraught with ethical dilemmas and risks such as immune rejection and tumorigenesis [62,63]. iPSCs, while circumventing ethical issues, still pose challenges related to tumor formation and genetic abnormalities [64,65]. MSCs are lauded for their immunomodulatory properties and growth factor secretion but face hurdles in cell survival at the injury site [66,67]. NSPCs, capable of generating neural cells and providing neurotrophic support, also grapple with survival and integration issues [68,69].

To assess the efficacy of these stem cell types, we have incorporated a summary of quantitative data from reputable studies. Metrics such as motor and sensory functional recovery, neural tissue preservation, and the reduction in gliosis and inflammation are considered. Standardized scales such as the Basso, Beattie, and Bresnahan (BBB) scale have been employed in both animal models and clinical trials to gauge locomotor recovery and sensory function [81,82]. Histological analyses provide quantitative measures of neural tissue preservation [83,84], and markers such as GFAP and cytokine levels are used to assess the reduction in gliosis and inflammation [85]. Comparative studies further enrich our understanding, indicating that each stem cell type has its unique set of advantages and limitations [86,87]. Additionally, different types of stem cells have been investigated in various models of SCI, including compression, contusion, section, and hemisection models. ESCs have been particularly effective in contusion models, showing promise in replacing lost neurons and remyelinating axons [88]. iPSCs have been found to be effective in section models, especially in promoting axonal regeneration [65]. MSCs have demonstrated efficacy across multiple models, notably in modulating immune responses and reducing inflammation [89]. NSPCs have been particularly effective in contusion models, where they have contributed to functional recovery and tissue repair [70]. Comparative studies indicate that the type of SCI model can influence the effectiveness of the stem cell type used [71]. Ongoing research aims to tailor stem cell therapies to specific injury types for optimal outcomes [72].

By synthesizing these data, we aim to offer a nuanced, evidence-based evaluation of the therapeutic efficacy of different stem cell types in SCI treatment. This approach not only enhances our understanding of the therapeutic potential of each stem cell type but also provides a robust foundation for future research in regenerative medicine for SCI.

### 3.1. Embryonic Stem Cells

ESCs are pluripotent cells derived from the inner cell mass of blastocysts and have the ability to differentiate into all cell types found in the adult body (Figure 1) [73]. This broad differentiation potential makes them an attractive option for SCI therapy, as they can potentially replace lost neurons [74,75], remyelinate axons [90,91], and facilitate tissue repair following injury [92]. However, the use of ESCs in research and therapy is associated with ethical concerns related to the destruction of embryos during the derivation process. These concerns have sparked debate and led to the implementation of strict guidelines and regulations surrounding ESC research in many countries [76]. Additionally, the use of ESCs for transplantation carries the potential risk of immune rejection, as these cells are allogeneic, meaning they are derived from a genetically different individual [77,78]. This can result in immune responses against the transplanted cells, limiting their therapeutic efficacy.

Another challenge associated with ESCs is the potential for tumor formation following transplantation [79]. Since ESCs are pluripotent, they have the capacity to form teratomas, which are tumors composed of a mixture of cell types derived from all three germ layers. This risk must be carefully managed in order to ensure the safety and efficacy of ESC-based therapies. Despite these challenges, researchers are continuing to investigate methods to harness the regenerative potential of ESCs while addressing the associated ethical, safety, and efficacy concerns. Advances in differentiation protocols, cell culture techniques, and transplantation strategies may help pave the way for the successful use of ESCs in SCI therapy [80,93].

### 3.2. Induced Pluripotent Stem Cells

iPSCs are generated by reprogramming adult somatic cells to a pluripotent state, circumventing the ethical concerns associated with ESCs (Figure 1) [94]. As with ESCs, iPSCs have the ability to differentiate into various cell types and have shown promise in preclinical studies for SCI treatment [95,96]. iPSC-derived neural cells have been used to replace lost neurons, promote axonal regeneration, and provide neurotrophic support, contributing to functional recovery in animal models of SCI [97,98]. However, iPSCs also carry a risk of tumor formation, as their pluripotent nature can lead to uncontrolled cell growth and differentiation [99,100]. Moreover, the reprogramming process may introduce genetic and epigenetic abnormalities that could affect the safety and efficacy of iPSC-based therapies. For instance, the use of viral vectors for reprogramming may result in insertional mutagenesis, which can disrupt normal gene function and increase the risk of tumorigenicity [101].

To address these challenges, researchers are exploring alternative reprogramming methods, such as using non-integrating viral vectors or small molecules, to generate safer iPSCs with reduced risks of genetic and epigenetic alterations [102]. Additionally, strategies to promote the directed differentiation of iPSCs into specific neural cell types and to enhance their integration into the host tissue are being developed to improve the therapeutic potential of iPSCs for SCI treatment [103,104]. Overall, iPSCs offer a promising alternative to ESCs in regenerative medicine for SCI, but further research is needed to overcome the safety concerns and optimize their use in therapy.

### 3.3. Mesenchymal Stem Cells

MSCs are multipotent cells found in various adult tissues, including bone marrow, adipose tissue, and umbilical cord blood (Figure 1) [105]. MSCs have demonstrated immunomodulatory properties and secrete a variety of growth factors, making them an attractive option for SCI therapy [106,107]. Their immunomodulatory properties help to suppress inflammation and modulate immune responses, which can be beneficial in minimizing secondary damage following SCI [108]. Furthermore, MSCs secrete a variety of growth factors and cytokines that promote angiogenesis, neurogenesis, and axonal growth, which contribute to tissue repair and functional recovery after SCI [109,110]. In preclinical studies, MSC transplantation has been shown to promote functional recovery and tissue repair after SCI, by reducing the lesion size, preserving myelin, and promoting axonal regeneration [111,112]. Additionally, MSCs are relatively easy to isolate and expand in vitro, making them a more accessible source of stem cells for therapeutic applications. They also have a lower risk of immune rejection due to their low expression of major histocompatibility complex molecules and immunosuppressive properties [113]. Moreover, MSCs have a lower risk of tumor formation compared to ESCs and iPSCs, as they are multipotent cells with a more limited differentiation potential [114].

Despite these advantages, there are some challenges associated with MSC-based therapies for SCI. One major challenge is the low survival rate of transplanted MSCs, which can be affected by the harsh microenvironment at the injury site [115]. Researchers are working to enhance MSC survival and engraftment by optimizing transplantation methods, preconditioning MSCs, or using biomaterial scaffolds to support cell survival [116,117,118]. Furthermore, the optimal source of MSCs, as well as the timing, dose, and route of administration, need to be determined to maximize the therapeutic potential of MSCs for SCI treatment. Overall, MSCs hold great promise for SCI therapy, but further research is required to address these challenges and optimize their use in clinical applications.

### 3.4. Neural Stem/Progenitor Cells

NSPCs can be derived from various sources, including embryonic and adult neural tissue, as well as iPSCs and ESCs (Figure 1) [119]. These cells have the inherent ability to generate neural cells, including neurons, astrocytes, and oligodendrocytes [120,121]. NSPC transplantation has shown promise in preclinical studies, promoting functional recovery, reducing lesion size, and enhancing remyelination in animal models of SCI [36]. Moreover, NSPCs can secrete neurotrophic factors that support the survival and regeneration of host neurons, and modulate the local environment to enhance endogenous repair processes [98]. The use of NSPCs may help overcome some of the challenges associated with other stem cell types, such as ethical concerns and risks of tumor formation [122,123]. As they can be derived from adult neural tissue or iPSCs, the ethical concerns associated with the use of ESCs can be circumvented. Furthermore, since NSPCs are committed to a neural lineage, they are less likely to form tumors compared to pluripotent stem cells such as ESCs and iPSCs [124,125]. However, there are still challenges that need to be addressed in the use of NSPCs for SCI treatment. The survival and integration of transplanted NSPCs into the host tissue can be limited by the harsh microenvironment at the injury site [126]. Strategies to improve cell survival and integration, such as preconditioning, biomaterial scaffolds, and the modulation of the injury microenvironment, are being explored to enhance the therapeutic potential of NSPCs [127,128]. Additionally, the optimal source of NSPCs, as well as the timing, dose, and route of administration, need to be determined for maximal therapeutic benefits.

### 3.5. Clinical Outcomes and Adverse Events

The application of stem cell transplantation in SCI has garnered significant attention due to its potential for tissue regeneration and functional recovery. However, the approach is not without challenges. This section aims to discuss three critical aspects of stem cell transplantation in SCI: the rate of immune rejection, the incidence of improvement and failure to improve, and the occurrence of cancer or other adverse events.

#### 3.5.1. Immune Rejection

The immune response to stem cell transplantation in SCI is complex and can vary depending on the type of stem cells used. Torres-Espín et al. (2013) found that transplantation of MSCs or OECs leads to an up-regulation of genes related to tissue protection and regeneration. However, there is also a marked increase in the expression of genes associated with foreign body response and adaptive immune response, suggesting a fast rejection response to the grafted cells [129]. This is consistent with other studies that have noted that the immune response in SCI can both exacerbate pathology and promote tissue repair [130].

#### 3.5.2. Rate/Incidence of Improvement and Failure to Improve

While the manuscript initially did not include specific rates of improvement or failure, it is crucial to note that stem cell transplantation has shown promise in various studies. For example, intravenous transplantation of bone marrow mesenchymal stem cells improved motor function in rats with SCI [66]. However, the effectiveness of the therapy can be influenced by the type of stem cells used and the timing of the transplantation [131].

#### 3.5.3. Occurrence of Cancer or Other Adverse Events

The risk of tumor formation is a significant concern, especially with pluripotent stem cells such as ESCs and iPSCs. However, MSCs and NSPCs generally have a lower risk of tumor formation due to their more limited differentiation potential.

#### 3.5.4. Other Adverse Events

The harsh microenvironment at the injury site can affect the survival rate of transplanted stem cells [129]. Moreover, SCI often leads to chronic inflammation and immune dysfunction, which can further complicate the transplantation process [132].

In summary, stem cell transplantation for SCIs holds immense promise for functional recovery, yet it comes with its own set of complex challenges, including immune rejection, inconsistent rates of improvement, and potential adverse events such as tumor formation. Each type of stem cell—ranging from NSPCs to MSCs—brings its own unique advantages and limitations to the table. A nuanced understanding of these characteristics is crucial for optimizing therapeutic strategies and making stem cell transplantation a viable clinical option. As we move forward, the focus of research should be on elucidating the mechanisms of action, refining transplantation protocols, and developing targeted therapies to mitigate these challenges. By doing so, we can pave the way for more effective and safer treatments that have the potential to significantly improve the quality of life for those affected by SCIs. Continued innovation and multidisciplinary research in this field are essential for realizing this potential and overcoming the existing hurdles.

## 4. Challenges in Stem Cell Therapy for SCI

Despite the tremendous potential of stem cell therapy for SCI treatment, the field still faces significant challenges (Figure 2). Ensuring the safety and efficacy of stem cell-based therapies is paramount. For example, the risk of tumor formation and the potential for immune rejection must be carefully addressed [133,134]. Furthermore, the development of standardized protocols for stem cell isolation, expansion, and differentiation is crucial for the successful translation of stem cell therapies from the laboratory to the clinic. In addition to safety concerns, another major challenge is the optimization of stem cell transplantation parameters, such as the timing of transplantation, the choice of cell type, and the most effective route of administration [135,136]. Research has shown that the timing of transplantation can have a significant impact on the therapeutic outcome, as the injured spinal cord undergoes various stages of inflammation, secondary injury, and tissue repair [137,138]. Some studies suggest that transplantation is most effective when carried out 1–2 weeks after injury [139], while others recommend a therapeutic window of 3–4 weeks following injury [140]. These timing considerations are crucial as the injured spinal cord undergoes various stages of inflammation, secondary injury, and tissue repair [137,138]. The choice of cell type is also a critical factor. For example, a study by Dai et al. (2013) confirmed that CT-guided stem cell transplantation is a safe and effective approach for treating sequelae of SCI, offering advantages such as minimal invasion and quicker recovery [141]. Identifying the optimal window for stem cell transplantation may be crucial for maximizing the benefits of stem cell therapy. The choice of cell type is also a critical factor in determining the success of stem cell therapy for SCI. As mentioned earlier, different stem cell types offer unique advantages and challenges [17,28]. Researchers need to determine which cell type or combination of cell types will provide the best therapeutic outcomes for SCI patients while minimizing the risks associated with each cell type.

Another challenge is the need for effective strategies to enhance the survival, integration, and function of transplanted stem cells within the host tissue [18,142]. This may involve the use of adjuvant therapies, such as growth factors, neurotrophic factors, or immunomodulatory agents, to promote cell survival and improve the local environment for stem cell integration and differentiation. Moreover, the development of suitable biomaterials and scaffolds that support cell survival, integration, and function is critical for the successful translation of stem cell therapies from the laboratory to the clinic [143,144]. There is also a need for the development of reliable and accurate outcome measures to evaluate the efficacy of stem cell therapies in clinical trials [145]. This may include the establishment of standardized functional assessments, imaging techniques, and biomarkers that can provide objective measures of treatment success and facilitate the comparison of results across different trials. Lastly, the cost and accessibility of stem cell therapies for SCI patients must be addressed. Developing effective, safe, and affordable stem cell therapies will be essential in ensuring that these treatments can be made widely available to those who need them.

In conclusion, we have highlighted the major challenges associated with stem cell therapy for SCI and proposed potential solutions to address these issues (Table 3). Overcoming these obstacles is crucial for successfully translating stem cell therapies from the laboratory to clinical settings and ultimately improving the lives of SCI patients.

### 4.1. Secondary Injury and Stem Cell Transplantation

The impact of secondary injury on stem cell transplantation in SCI is a multifaceted topic that warrants in-depth exploration. This section aims to shed light on how secondary injury mechanisms interact with stem cell transplantation, affecting outcomes such as immune rejection, tissue repair, and overall efficacy.

#### 4.1.1. Prevention of Secondary Damage

Grafted neuroectodermal stem cells have been shown to prevent secondary spinal cord damage and induce significant regeneration through multiple mechanisms [146].

#### 4.1.2. Restriction of Tissue Damage

Resident neural stem cells are required to restrict secondary enlargement of the lesion and further axonal loss after SCI [147].

#### 4.1.3. Optimal Timing

A precise window of opportunity exists for the treatment of complex SCIs with therapeutically plastic somatic stem cells [70].

#### 4.1.4. Microenvironment

The harsh microenvironment at the injury site can affect the survival rate of transplanted stem cells, which is a challenge that needs to be addressed for effective therapy [129].

Understanding the interplay between secondary injury and stem cell transplantation is crucial for optimizing therapeutic strategies in SCI. While stem cells have the potential to mitigate secondary damage and promote tissue repair, challenges such as immune rejection and the harsh microenvironment at the injury site remain. Future research should focus on identifying the optimal timing and conditions for stem cell transplantation to maximize its therapeutic efficacy.

## 5. Future Directions and Technological Advancements

As the field of stem cell therapy for SCIs continues to evolve, technological advancements are paving the way for more effective treatments. One of the most promising areas of research is the use of gene editing technologies such as CRISPR/Cas9 to modify stem cells for improved therapeutic efficacy, safety, and immunocompatibility [148,149,150].

### 5.1. Interdisciplinary Approaches to Advancing Stem Cell Therapies for SCIs

#### 5.1.1. Biomaterials and Differentiation

Biomaterials play a significant role in stem cell therapy for SCIs. Advanced scaffolds and biomaterials that mimic the native extracellular matrix can provide a conducive environment for stem cell integration, survival, and differentiation [127,151]. For instance, 3D-printed biomimetic spinal cords have been shown to promote directional differentiation and repair motor function after SCI [152]. Moreover, the mechanical properties of scaffolds can direct stem cell differentiation in vivo [153].

#### 5.1.2. Cell Engineering Techniques

Gene editing techniques such as CRISPR/Cas9 can be used to improve the therapeutic potential of stem cells. These modifications can enhance safety and immunocompatibility, although further research is needed to confirm their efficacy in SCIs [87].

#### 5.1.3. Clinical Translations

Several studies have demonstrated the potential of stem cell therapies in clinical settings. For example, the transplantation of human umbilical cord-mesenchymal stem cells on a collagen scaffold has been shown to promote the recovery of neurological function after acute SCI [154].

#### 5.1.4. Challenges and Solutions

Despite these advancements, challenges such as safety, efficacy, and standardization of protocols remain. Research indicates that the optimal timing for transplantation is crucial, and various approaches such as the pharmacological activation of endogenous stem cells and optogenetics are being explored to improve the application of stem cell therapy [87].

#### 5.1.5. Quality of Life Improvements in SCI Patients

Clinical studies have shown that stem cell transplantation can significantly improve the quality of life for patients with SCI. For example, a study involving six patients with chronic SCI found that autologous co-transplantation of bone marrow mesenchymal stem cells and Schwann cells led to improvements in bladder compliance and axonal regeneration, with no adverse findings over a mean 30 months of follow-up [50]. Another study demonstrated that CT-guided stem cell transplantation was a safe and effective approach for treating sequelae of SCI, offering advantages such as minimal invasion and quicker recovery [141]. Moreover, transplantation of human spinal cord-derived neural stem cells has been shown to improve motor or sensory function in individuals with chronic SCI [49].

In summary, advancements in stem cell therapy for SCIs are accelerating at an unprecedented rate, fueled by technological innovations and multidisciplinary research. The synergy between biomaterial science, genetic engineering, and clinical studies is paving the way for more effective and personalized treatments. However, challenges such as safety protocols, efficacy validation, and treatment standardization remain. As we navigate these complexities, collaborative initiatives across scientific, clinical, and industrial sectors become increasingly vital. These partnerships will be instrumental in overcoming obstacles and unlocking the full therapeutic potential of stem cell treatments for SCIs.

### 5.2. Challenges and Problems in Non-Traumatic SCI

The application of stem cells for treating SCIs resulting from diseases such as transverse myelitis and neuromyelitis optica presents unique challenges and opportunities. While the underlying pathology may differ from traumatic SCIs, the therapeutic potential and limitations of stem cell interventions remain critical areas of investigation. This section delves into the multifaceted issues surrounding the use of stem cells in non-traumatic SCIs, including safety, timing, microenvironment, ethical concerns, and efficacy.

#### 5.2.1. Safety and Feasibility

Clinical trials have shown the safety of autologous co-transplantation of bone marrow mesenchymal stem cells and Schwann cells, but long-term follow-up is essential [50].

#### 5.2.2. Optimal Timing

A precise window of opportunity exists for stem cell transplantation, which varies depending on the type and severity of the injury [70].

#### 5.2.3. Microenvironment

The harsh microenvironment at the injury site can affect the survival rate of transplanted stem cells [147].

#### 5.2.4. Ethical Concerns

The source of stem cells can raise ethical issues, although some types such as unrestricted somatic stem cells lack such concerns [155].

#### 5.2.5. Efficacy

The effectiveness of stem cell therapies in non-traumatic SCI is still under investigation, and premature application should be discouraged [156].

The challenges in applying stem cells for non-traumatic SCIs are complex and require a multi-faceted approach. While there is promising evidence for the safety and potential efficacy of such treatments, long-term studies are essential. Ethical considerations and the harsh microenvironment at the injury site further complicate the therapeutic landscape. Future research should focus on addressing these challenges to expand the applicability and effectiveness of stem cell therapies in non-traumatic SCIs.

### 5.3. The Role of Biomaterials, Engineering, and Collaborative Research for SCI Therapies

Biomaterials and tissue engineering techniques can also play a crucial role in the development of more effective stem cell therapies. By designing advanced scaffolds and biomaterials that mimic the native extracellular matrix, researchers can provide a more conducive environment for stem cell integration, survival, and differentiation [127,151]. Moreover, these materials can also be tailored to release growth factors or other bioactive molecules that can further enhance the regenerative process. One emerging approach that holds promise for SCI treatment is the combination of stem cell therapy with electrical stimulation or rehabilitation strategies. The synergistic effects of stem cell transplantation and rehabilitation may lead to improved functional recovery by promoting neural plasticity and enhancing the integration of transplanted cells into the host tissue [157,158]. Further research in this area could lead to the development of more effective and personalized therapeutic interventions for SCI patients. Collaborative efforts between basic scientists, clinicians, engineers, and industry partners will be essential to address the multifaceted challenges in translating stem cell therapies from the laboratory to clinical applications. Additionally, the development of international consortia, multicenter clinical trials, and standardized outcome measures will be crucial in generating high-quality evidence to support the widespread adoption of stem cell therapies for SCIs.

In summary, stem cell therapy for SCIs represents a beacon of hope amid the numerous challenges that need to be addressed. As researchers continue to make strides in overcoming these obstacles and refining stem cell therapies, we can anticipate groundbreaking advancements that will revolutionize the way we approach SCI treatment. By working together to tackle the challenges and address ethical concerns, the potential of stem cell therapy to improve the lives of those affected by SCIs can be fully realized.

## 6. Conclusions

Stem cell therapy holds promise for SCI treatment by potentially revolutionizing damaged neural tissue repair and regeneration. As research progresses and challenges are addressed, stem cell therapy could significantly impact the lives of those affected by SCIs, offering them hope for a better future. Collaborative efforts among scientists, clinicians, bioengineers, and industry partners are essential for driving progress in stem cell therapies for SCIs. By exploring various stem cell types, developing standardized protocols, and incorporating advancements in gene editing, biomaterials, and tissue engineering, researchers may develop more effective interventions and personalized therapeutic strategies. Furthermore, continued investment in stem cell research and the establishment of international collaborations and networks will facilitate the exchange of knowledge and resources, enabling rapid advancements in the field. Addressing ethical concerns, ensuring regulatory compliance, and fostering public awareness and understanding of stem cell therapies are also crucial to garner support for these potentially life-changing treatments.

In addition to the scientific and technical challenges, clinical trials and long-term follow-up studies will be necessary to evaluate the safety and efficacy of stem cell therapies in the context of SCI treatment. Patient advocacy and involvement will play a crucial role in ensuring that new therapies are developed with the best interests of SCI patients in mind, taking into account their specific needs and desires. Ultimately, realizing the full potential of stem cell therapy for SCI treatment relies on the scientific community’s dedication, passion, and persistence. By working together and pushing the boundaries of our knowledge, we can provide hope and an improved quality of life for countless individuals affected by SCIs, transforming the landscape of SCI treatment for generations to come.

## Figures and Tables

**Figure 1 ijms-24-14349-f001:**
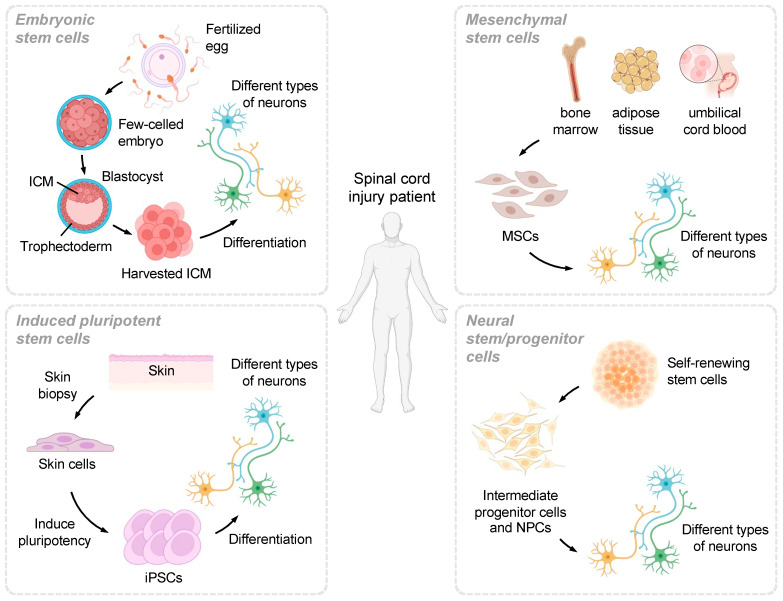
Overview of various stem cell types explored for their potential in SCI treatment. This figure provides a visual summary of the four major stem cell types discussed in the review article: embryonic stem cells (ESCs), induced pluripotent stem cells (iPSCs), mesenchymal stem cells (MSCs), and neural stem/progenitor cells (NSPCs). Each stem cell type is represented by a separate section in the figure, highlighting their unique advantages and potential differentiation into different types of neural cells for SCI therapy. The figure aims to facilitate understanding of the distinct characteristics and capabilities of these stem cell types, enabling researchers to optimize their use in regenerative medicine applications for SCI treatment.

**Figure 2 ijms-24-14349-f002:**
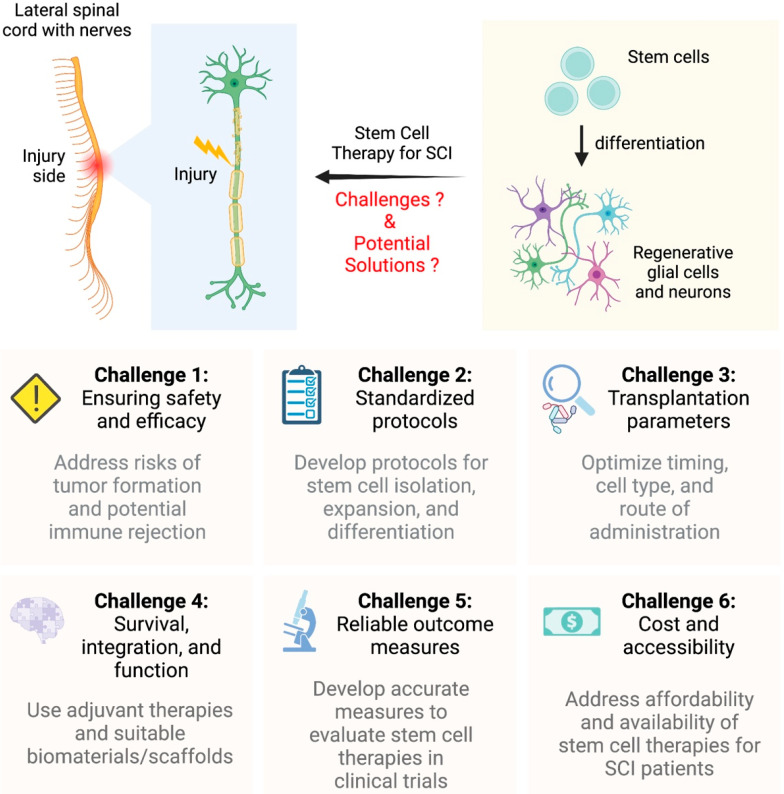
Overview of challenges and potential solutions in stem cell therapy for SCIs. This infographic illustrates the main challenges and potential solutions in the development and implementation of stem cell therapies for SCI treatment. Six key challenges are represented in hexagons surrounding a central image of a spinal cord with an overlaying stem cell icon, symbolizing the application of stem cell therapy for SCI. Each hexagon contains an icon representing the challenge, along with a brief description of the corresponding solution. The challenges and solutions addressed in the figure include ensuring safety and efficacy, developing standardized protocols, optimizing transplantation parameters, enhancing survival, integration, and function of transplanted stem cells, establishing reliable outcome measures, and addressing cost and accessibility of stem cell therapies for SCI patients.

**Table 1 ijms-24-14349-t001:** Epidemiological data on SCIs: incidence, prevalence, and demographics.

	Data	Country/Region	Age Group Most Affected	Primary Causes of Injury	Notable Trends	Year	Reference
Incidence Rates	32–50 per million requiring hospital admission	United States	N/A	N/A	N/A	2000	[5]
	11.4 per million	Spain	60–69 years	Tumors	N/A	2012	[6]
	19.4 per million inhabitants	France	N/A	N/A	N/A	2005	[7]
	Stable incidence rate, changes in injury etiology	Stockholm, Sweden	N/A	Falls, Transport-related	N/A	2017	[8]
	23 cases per 1,000,000 persons	Global	N/A	N/A	N/A	2007	[9]
Prevalence	Estimate provided, further refinements needed	Victoria, Australia	N/A	N/A	N/A	2012	[10]
Demographic Trends	Increased incidence due to falls and motor vehicle accidents	Global	N/A	Falls, Motor Accidents	Increasing	2010	[11]
	Increased incidence in the elderly	Global	Elderly	Falls, Non-traumatic	Increasing	2010	[12]
	Mean age at injury increased	Finland	N/A	N/A	Increasing	2008	[13]
	Specific epidemiological characteristics	Xi’an, China	N/A	N/A	N/A	2020	[14]
Global Trends	Significant variation in incidence worldwide	Global	N/A	N/A	Varies	2010	[11]
	Variation in etiology, male-to-female ratios, age distributions, and complications	Global	N/A	N/A	Varies	2004	[15]

Note: Some fields are marked as N/A (Not Available) due to the absence of specific information in the cited studies.

**Table 2 ijms-24-14349-t002:** Comparison of stem cell types for SCI treatment.

Stem Cell Type	Advantages	Challenges	References
Embryonic Stem Cells (ESCs)	- Pluripotent - Can differentiate into all adult cell types	- Ethical concerns - Risk of immune rejection - Risk of tumor formation	[52,56,57,58]
Induced Pluripotent Stem Cells (iPSCs)	- Pluripotent - Ethical concerns circumvented - Can differentiate into various cell types	- Risk of tumor formation - Potential genetic and epigenetic abnormalities	[66,67,68,69]
Mesenchymal Stem Cells (MSCs)	- Multipotent - Immunomodulatory properties - Secrete growth factors - Easy to isolate and expand - Lower risk of immune rejection and tumor formation	- Limited differentiation potential compared to ESCs and iPSCs	[70,71,72,73,74,75]
Neural Stem/Progenitor Cells (NSPCs)	- Inherent ability to generate neural cells - Promote functional recovery and reduce lesion size in preclinical studies - Overcome ethical concerns and risks of tumor formation	- Source-dependent properties	[76,77,78,79,80]

**Table 3 ijms-24-14349-t003:** Stem cell types, advantages, existing challenges, and possible solutions in stem cell therapy for SCI.

Stem Cell Type	Advantages	Existing Challenges	Possible Solutions
Embryonic Stem Cells (ESCs)	- Pluripotent - Can differentiate into all adult cell types	- Ethical concerns - Immune rejection - Tumorigenic potential	- Utilize alternative stem cell sources - Improve immunocompatibility - Rigorous safety testing
Induced Pluripotent Stem Cells (iPSCs)	- Pluripotent - Ethical concerns circumvented - Can differentiate into various cell types	- Genetic and epigenetic aberrations - Tumorigenic potential - Variability in differentiation capacity	- Refine reprogramming techniques - Standardize differentiation protocols
Mesenchymal Stem Cells (MSCs)	- Multipotent - Immunomodulatory properties - Secrete growth factors - Easy to isolate and expand - Lower risk of immune rejection and tumor formation	- Limited ability to differentiate into specific neuronal subtypes - Inconsistent therapeutic outcomes	- Enhance differentiation capacity - Combine with other stem cell types - Improve standardization and optimization
Neural Stem/Progenitor Cells (NSPCs)	- Inherent ability to generate neural cells - Promote functional recovery and reduce lesion size in preclinical studies - Overcome ethical concerns and risks of tumor formation - Can integrate into host tissue	- Limited availability - Invasive extraction	- Develop strategies for expansion - Explore alternative sources

## Data Availability

Not applicable.

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
