# Peer review of "Advancing Spinal Cord Injury Treatment through Stem Cell Therapy: A Comprehensive Review of Cell Types, Challenges, and Emerging Technologies in Regenerative Medicine"

_ijms, 2023, doi:10.3390/ijms241814349_

Round 1

Reviewer 1 Report

The review presented by the author is very interesting, however, in order to improve the manuscript, several changes must be introduced in the final version:

INTRODUCTION

Updated data on the incidence and prevalence of spinal cord injury should be included in the introduction of the final version of the manuscript.

What new information does this review provide compared to previous reviews on the same topic? What new approach does this review bring with respect to previous reviews already published on the same topic? The novelty of the review must be included in the final version of the manuscript.

Why is it important to return to another review on stem cells and their applicability to spinal cord injury? Include this information in the final version of the manuscript.

THE PROMISE OF STEM CELL THERAPY FOR SCIS

Most preclinical studies show that stem cells transplanted into spinal cord injury are 90% differentiated into glial cells, and only 10% differentiated into new neurons, which in many cases do not integrate into the cells. established neural circuits. Why does the author indicate that this therapy is very promising for spinal cord injuries? Please clarify this point and include this information in the final version of the manuscript.

How many clinical studies have been carried out in patients with spinal cord injury who have received a stem cell transplant, and that this cell therapy has been successful, allowing the recovery of motor, sensory and autonomic nervous system functions lost after the injury? The author must include this information in the final version of the manuscript.

What are the advantages and disadvantages of stem cell transplantation compared to transplantation of other types of cells such as Schwann cells, olfactory ensheathing glia cells, etc.? The author must include this comparative information between stem cells and non-stem cells in the final version of the manuscript.

THE POTENTIAL OF DIFFERENT STEM CELL TYPES

The author must include quantitative data from preclinical and clinical studies on the degree of motor and sensory functional recovery, as well as the degree of preservation of neural tissue, or reduction of gliosis and inflammation, in the final version of the manuscript. This information about each of the types of stem cells is very relevant, to understand the success of this therapy. Please include this information in the final version of the manuscript.

What type of stem cell is most appropriate to improve spinal cord injury functionally and histologically? Are there comparative studies on the efficacy and efficiency of transplants of different types of stem cells in spinal cord injury? Please include this information in the final version of the manuscript.

In which model of spinal cord injury (e.g., compression, contusion, section, hemisection) these different types of stem cells have been implanted? Which of these different types of stem cells have been best at regaining lost function based on the type of spinal cord injury? Please include all this information in the final version of the manuscript.

CHALLENGES IN STEM CELL THERAPY FOR SCI

The author should treat this section in greater depth, giving more quantitative and less qualitative data.

FUTURE DIRECTIONS AND TECHNOLOGICAL ADVANCEMENTS

In this section the author must include specific studies and give more precise information on the different points that he develops in the section. How do different biomaterials influence the differentiation and integration of stem cells transplanted in spinal cord injury? What improvements have been made in stem cells using cell engineering techniques? Have these modified stem cells been successful in spinal cord injury? What have been the main findings of stem cell transplantation that have allowed its use in the clinic? These are some of the questions that the author can answer in depth in this section of the manuscript. Please include all this new and relevant information in the final version of the manuscript.

How does stem cell transplantation improve the quality of life of patients with spinal cord injury? The author must include information on clinical studies explaining in depth the most relevant changes that have allowed a better quality of life to be given to patients who have received this cell therapy. This information is very relevant and should be included in the final version of the manuscript.

GENERAL CONSIDERATIONS

In general, this review lacks more quantitative evidence, explaining more and better the multitude of preclinical studies in which stem cells have been used in spinal cord injury, better explaining the concentration of implanted cells, the type of injury, the animal species, the level of injury, the transplantation time from the spinal cord injury, and the success of motor and sensory recovery after transplantation. All these preclinical data are important, and the author must try to include them in the final version of the manuscript. It is also very important to add quantitative data from clinical studies on stem cell transplants in patients with spinal cord injury, indicating the degree of injury to the patient (ASIA scale), the type of stem cell transplanted and its concentration, the level of injury, and the degree of post-transplant functional recovery. All this information should be included in the final version of the manuscript.

Reviewer 2 Report

The authors write a coherent and informed high-level review of the challenges of stem cell therapy for spinal cord injury.  While the high-level identification of problems/challenges and corresponding pictorial diagrams are very nice, the review lacks critical scientific details and substance that will make it impactful to current leaders in the field or adjacent domains (bioengineering, machine learning, biomaterials, etc.) that have tools that could be utilized for the stem cell applications in SCI.

The authors have repeated many of their high-level ideas across the text, figures, and tables. Some of this duplicity could be removed without altering the very nice coherent structure and framing.  Most importantly, the authors need to add more details -  specifically what stem cell lines, models (experimental and clinical), protocols, outcome measures, etc. have been implemented and what were their quantitative results?  What is the rate of immune rejection, rate/incidence of improvement, rate/incidence of failure to improve, occurrence of cancer or other adverse events due to inappropriate stem cell growth, etc.?  Specifically what types of biomaterials might make better environments for holding 3-D tissue engineered constructs of stem cells?  Can specific types of machine learning assist with identifying personalized biomarkers or personalized stem cell constructs? What do state-of-the art studies specifically say on the timing of potential stem cell introduction?  What are the pros and cons of early introduction versus late introduction, and how are "early" and "late" defined?  Exactly how does secondary injury help or hinder the introduction of stem cells and what are the overlapping mechanisms/etiology that are modulated? These are just examples of a plethora of more details that could enhance the paper.  

In summary, details, details, details, are what is necessary to make this a valuable, impactful work that goes beyond the high-level ideas that are predominantly already known by those researchers who are in the domain. Supporting the review with more aggregated quantitative data - like a meta-analysis with aggregate effect sizes - would be another way to greatly improve the work.  Minimally, the authors need to provide tabulated quantitative data to support the higher level ideas they have already nicely constructed with appropriate scientific art.

Finally, while this review is focused on traumatic injury SCI (injury due to mechanical trauma and related secondary injury), one paragraph on the application of stem cells and corresponding problems/challenges due to spinal cord injury from disease like transverse myelitis, neuromyelitis optica, etc. could further expand the readership and corresponding impact.  Many of the problems/challenges are similar and a discussion of this intersection would be of high impact to the field.

Round 2

Reviewer 1 Report

The author has made an effort to respond to the reviewer's suggestions, and has included most of the new information in the final version of the manuscript.

Reviewer 2 Report

The authors have greatly improved the paper.